# Solid Particle Erosion of Filled and Unfilled Epoxy Resin at Room and Elevated Temperatures

**DOI:** 10.3390/polym15010001

**Published:** 2022-12-20

**Authors:** Maxim Mishnev, Alexander Korolev, Dmitrii Ulrikh, Anna Gorechneva, Denis Sadretdinov, Danila Grinkevich

**Affiliations:** 1Department of Building Construction and Structures, South Ural State University, Chelyabinsk 454080, Russia; 2Department of Town Planning, Engineering Systems and Networks, South Ural State University, Chelyabinsk 454080, Russia

**Keywords:** solid particle erosion, epoxy resin, thermoset polymers, fly-ash, filled compound, elasticity modulus, bending strength

## Abstract

Solid particle erosion at room and elevated temperatures of filled and unfilled hot-cured epoxy resin using an anhydride hardener were experimentally tested using an accelerated method on a special bench. Micro-sized dispersed industrial wastes were used as fillers: fly ash from a power plant and spent filling material from a copper mining and processing plant. The results showed that the wear of unfilled epoxy resin significantly decreases with increasing temperature, while the dependence on the temperature of the wear intensity at an impingement angle of 45° is linear and inversely proportional, and at an angle of 90°, non-linear. The decrease in wear intensity is probably due to an increase in the fracture limit because of heating. Solid particle erosion of the filled epoxy compounds is considerably higher than that of unfilled compounds at impingement angles of 45° and 90°. Filled compounds showed ambiguous dependences of the intensity of wear on temperature (especially at an impingement angle of 45°), probably as the dependence is defined by the filler share and the structural features of the samples caused by the distribution of filler particles. The intensity of the wear of the compounds at impingement angles of 45° and 90° has a direct and strong correlation with the density and the modulus of elasticity, and a weak correlation with the bending strength of the materials. The data set for determining the correlation between the mechanical properties and the wear included compound filling characteristics and temperature.

## 1. Introduction

Composite materials based on thermosetting resins (including epoxy resins) are widely used in various industries due to their corrosion resistance, high strength, and low weight. For example, they perform well in large shell constructions for industrial gas exhaust ducts [1,2,3,4], operating at temperatures up to 120 °C with low dust-particle content in the exhaust gases and, consequently, a low abrasive impact of the gases on the inner surface of the duct. Examples of such structures are shown in Figure A1 of Appendix C.

In the future is promising to use of structures made of polymeric composite materials for gas exhaust ducts (such as chimneys) of industrial enterprises with gas temperatures of up to 180 °C and the big content of ash particles and metallurgical dust, causing abrasive wear of the inner surface of structures (even steel) [5]. For example, such conditions are often inherent in the gas exhaust ducts of metallurgical enterprises or coal-fired thermal power plants. For practical implementation, it is necessary to predict the service life of such structures made of polymeric composites under the abrasive impact of solid particles present in flue gases at high temperatures. Therefore, research into resistance to the gas abrasive wear of polymeric materials and composites including at high temperatures has practical importance.

The problem of reducing the cost of composite materials used in the construction industry can be achieved by reducing the amount of expensive polymeric binder, by filling it with dispersed additives. Besides a decrease in cost, filling the binder may increase its mechanical characteristics (mainly stiffness) [6,7]. Different dispersed additives can be used as fillers [6,7,8] and an effective solution is the use of fine-dispersed industrial waste (ashes, slags, etc.), because of their low cost, as well as the need for their utilization.

In the present work, the object of research was filled and unfilled hot-cured epoxy binders on an anhydride hardener. The epoxy resin used in this study (KER 828) is a bisphenol A-based resin and is one of the most widespread and cost-effective, having many analogues produced in different countries under different brand names: ED-20 (Russia), NPEL128S (Taiwan), YD128S (South Korea), etc. [9]. Fine-dispersed industrial wastes were used as fillers: fly ash from SDPP and spent material from a copper mining and processing plant. The subject of the research was solid particle erosion at room and elevated temperatures (up to 180 °C). 

The main advantage of using dispersed additives as a filler compared with other types of additives (for example, with alumina, which is highly effective in combination with epoxy resin [10,11,12]) is their lower cost since they are by-products of industrial processes. The reuse of solid industrial waste also contributes to solving the ecological problem of their disposal.

Although unfilled polymeric binders are not structural materials themselves, we chose unreinforced epoxy binders as an object of study because the resistance to solid particle erosion of reinforced composites is lower than that of unreinforced polymeric binders used as a matrix [13,14,15,16]. The inner chemically-resistant layer [17], which is a non-reinforced polymer binder, is often used in the gas exhaust ducts, which are subject to abrasion, playing the role of a protective cover for the composite layer that takes the mechanical load.

The gas- and hydro-abrasive wear of polymeric materials and composites has been widely studied. In most studies, composites with filled and unfilled polymer matrices reinforced with different types of fibers are considered.

A well-known work on the study of solid particle erosion of polymer composites is [18]. It lists the main factors influencing the intensity of erosion and studies composites with different types of reinforcing fibers and an epoxy matrix. The wear of polymer composites is considerably higher than that of carbon steel at the same particle rate, and for ductile materials, the maximum wear occurs at an impingement angle of about 20°, and for brittle materials about 90°.

A review of factors influencing the intensity of solid particle erosion of glass-reinforced plastics with a filled polymer matrix is given in [8] based on literature data. The influence of fillers (artificial and natural) and experimental conditions (velocity, particle shape, impingement angle, etc.) is discussed. All the results were obtained at room temperature and in some cases, filling the polymer matrix improves the resistance of the reinforced composite to solid particle erosion.

The resistance to solid particle erosion of carbon-fiber reinforced composites with unfilled and graphite-powder filled epoxy matrix was studied at room temperature (the filler content by weight was 2–6% of the weight of the whole composite or 5–15% of the matrix weight) in [13]. The samples with an unfilled matrix showed higher wear resistance than the filled ones at all impingement angles, the most intensive wear was observed at an angle of 45°.

Solid particle erosion of epoxy resins modified with hydrothermally decomposed polyester-urethane was studied at room temperature, and corundum particles with sizes from 60 to 120 microns were used as an abrasive [19]. The modified binders had a higher wear resistance compared to the unmodified ones.

In [20] solid particle erosion of epoxy resin modified with synthetic oil was investigated at room temperature, and quartz sand was used as an abrasive, emitted at speeds from 6.5 to 9.5 m/s at different angles. The modified binders had less brittleness and higher wear resistance compared to the non-modified ones.

In [21] the solid particle erosion of epoxy fiberglass plastic at room temperature was investigated including filling of the matrix with fine-dispersed tungsten carbide powder. The filling improved the wear resistance of the fiberglass plastic, the microphotographs obtained using a scanning electron microscope showed more significant damage to the matrix and fibers in the composite with an unfilled matrix, and the most intense wear was observed at an impingement angle of 90°.

In [22], the solid particle erosion of epoxy resins filled with cenospheres were investigated at room temperature, and the result showed that filling the epoxy resin with cenospheres increased the wear resistance compared to the unfilled resin at all the impingement angles considered.

In [23], the solid particle erosion of glass-fiber reinforced plastic with epoxy matrix filled with fly ash from the Obra thermal power plant, Mirzapur, India was investigated at room temperature. The filled matrix reduced the wear rate of the fiberglass compared to the unfilled matrix. Hot-cured CY-205 epoxy resin with HY-951 anhydride hardener was used as the matrix.

In contrast to most of the above papers, [24], investigating solid particle erosion of carbon-fiber reinforced plastic with an epoxy matrix, unfilled and filled with nanoparticles montmorillonite, showed that the composite with unfilled epoxy matrix had the best wear resistance, that is, in some cases fillers can reduce the wear resistance of polymer composites.

In all these works, the test methods were similar, but differed in several parameters, such as abrasive particle velocity, distance to the sample, abrasive material shape, etc. All tests were conducted at room temperature.

These studies show that the influence of filling polymer thermosetting binders on their wear resistance under gas abrasion is unambiguous. However, there are no studies of the solid particle erosion of unfilled hot-cured epoxy resins filled with fly ash or copper-mining slag at temperatures up to 180 °C (which exceeds the glass transition temperature of most epoxy binders) in the literature. Such data would be useful for the prediction of solid particle erosion during the long-term operation of polymer composite structures, such as the gas exhaust ducts of metallurgical enterprises.

## 2. Materials and Methods

### 2.1. Materials

To obtain the compounds used in this work the following materials were used:

- KER 828 epoxy resin: epoxy group content (EGC) 5308 mmol/kg, equivalent epoxy weight (EEW) 188.5 g/eq, viscosity at 25 °C 12.7 Pa×s, HCl 116 mg/kg, total chlorine 1011 mg/kg. Manufacturer: KUMHO P&B Chemicals, Seoul, Korea.

- Isomethyltetrahydrophthalic anhydride (IZOMTGFA) (hardener for epoxy resin): viscosity at 25 °C 63 Pa×s, anhydride content 42.4%, volatile fraction content 0.55%, free acid 0.1%. Manufacturer: ASAMBLY Chemicals Company Ltd., Nanjing, China.

- Alkophen (epoxy curing booster): viscosity at 25 °C 150 Pa×s, molecular formula C15H27N3O, molecular weight 265, amine number 600 mg KOH/g. Manufacturer: JSC “Epital”, Moscow, Russian Federation.

- Fly ash from SDPP (Figure 1a): fraction 0.01–0.05 mm, specific surface 310 m^2^/kg, bulk density 700 kg/m^3^, the density of particles 1700 kg/m^3^, modulus of elasticity of particles (approximately) 20,000 MPa, composition—CaO, MgO, SO_3_, Na_2_O, K_2_O.

- Backfill material was obtained from mineral processing waste of the copper-mining combine (Figure 1b): it contains Quartz low (SiO_2_) 20–30%, Aluminum Magnesium Hydroxide Silicate 5–15%, Magnesium Silicate (Serpentine) MgSiO_3_ 5–10%, clay minerals (hydrated aluminosilicates) 20–40%. The material was pre-dried at 105°C and then crushed in a laboratory mill to a fraction of no more than 0.05 mm.

The compounds are shown in Table 1. The binders were mixed using a Stegler DG-360 mechanical disperser-homogenizer (China) with an M-shaped nozzle of diameter 17 mm and a nozzle rotation speed of 6000 rpm. After mixing, the binders were poured into silicone molds and placed in a laboratory oven for curing. The samples were cured at 110 °C for 30 min. After initial curing, all samples were incubated at 150 °C for 12 hours. The sample preparation process is shown in Figure 2.

All experimental samples were made from a single binder mixture and five batches of samples were made: 

- Batch №1 (EP1): immediately after mixing all the components (resin, hardener, and booster), the binder was poured into a silicone mold and placed in a laboratory oven for curing.

- Batch №2 (EP2): after mixing all the components, the binder was kept for one hour at room temperature then poured into a silicone mold and placed into a laboratory oven for curing.

- Batch №3 (EP-FA25): after mixing all components (including the filler), the binder was kept for an hour and a half at room temperature and then poured into a silicone mold and placed in a laboratory oven for curing.

- Batch №4 (EP-FA50): after mixing all components (including the filler), the binder was kept for two hours at room temperature and then poured into a silicone mold and placed in a laboratory oven for curing.

- Batch №5 (EP-FM33): after mixing all components (including the filler), the binder was kept for two and a half hours at room temperature and then poured into a silicone mold and placed in a laboratory oven for curing.

From each batch, 90° and 45° test samples were made (the same samples were tested consecutively at different temperatures, but at the same angle of impingement).

### 2.2. Methods

In addition to solid particle erosion tests, three-point bending tests were carried out to determine the modulus of elasticity and bending strength; the density of the samples was determined by hydrostatic weighing.

#### 2.2.1. Solid Particle Erosion Tests at Elevated Temperatures

To perform accelerated tests for solid particle erosion resistance at elevated temperatures, an experimental bench was designed and manufactured, described in detail in [16]. For the present work, it was additionally insulated from the outside since in the previous tests it was necessary to wait too long for heating to the temperature of 180–200 °C, Figure 3 shows the test bench in the insulated cladding.

The test bench ASTM compliant [25] was designed for accelerated gas-abrasion testing of lamellar samples of polymer and composite materials at room and elevated temperatures. The width of the tested samples was 40–50 mm, height 50–80 mm, thickness 1–10 mm.

The abrasive impact was carried out with the Metabo SSP 1000 standard blast pistol with a 6 mm exit diameter, a capacity of 300 L/min (5 m^3^/s), and an exit velocity of 115 m/s. The blast pistol can be set in two positions to test at angles of attack of 90° and 45°. The spent abrasive was poured into the lower part of the enclosure from where it was retrieved when the lower enclosure lid was opened.

The plate sample was clamped in front of the nozzle of the sandblasting gun and abraded. The distance from the end of the nozzle to the sample is 115 mm excluding the thickness of the sample. The diameter of the blasted spot after one test was 27–30 mm.

Inside the case, the air was heated by two heating elements of 1 kW each, the heating was controlled by the thermoregulator OVEN TRM500 with thermocouple DTPL054 00 100, installed on the upper cover of the case. Since in this design the compressed air was not heated, the abrasive material should be preheated to reduce the cooling of the sample during high-temperature tests.

Compressed air was provided to the sandblaster by the NEXTTOOL compressor KMK-2300/100V with a capacity of 420 L/min, maximum pressure of 8 bar, and volume of the vessel of 100 L. The pressure in the receiver at the beginning of the test was 7.5 bar and at the end of the test was 6.0 bar.

The dimensions of plate samples for wear tests were 60 × 40 mm, the initial thickness of samples was 4–8 mm. The dimensions of bend test samples: length 70 mm, width 10 mm (±1.0 mm), thickness 4 to 5 mm. Thickness of samples was different as they were mechanically processed (grinded) to remove surface defects. The actual dimensions of the tested samples were measured with a caliper with an accuracy of 0.01 mm.

Copper slag (Figure 4) with sharp angular-shaped grains of 0.125–0.63 mm in size, with a Mohs hardness of at least 6 and a grain density of 3.2–3.9 g/cm^3^ was used as an abrasive.

Before the erosion tests, the density of the samples was determined by hydrostatic weighing. Then, after drying, the plate sample was weighed with the accuracy of 0.001 g and placed in the clamps of the test bench. 600 g of abrasive powder (copper slag) was poured into the abrasive container of the sandblasting gun and the sample was treated for the 1st minute. After blasting the volume was measured and the spent abrasive weighed, the average volume was about the same and was 500 cm^3^ at a weight of 975–1025 g (average bulk density 1.95–2.05 g/cm^3^). The sample was then removed from the bench clamps and re-weighed, and the mass loss was determined. Through the loss of mass and density, the loss of material volume as a result of gas-abrasion erosion was calculated.

The temperature of the sample was monitored by an external thermocouple placed on the back and not exposed to abrasion. Since the test procedure involves cooling the sample through a stream of unheated air, the average between the pre-test and post-test temperatures was used as the temperature at which the wear was determined.

Because of the high intensity of the abrasive effect, the wear of samples will differ in a significant way from the wear of real constructions. However, the results of the tests can allow a comparison of the wear resistance of the materials used and, with some assumptions, predict the solid particle erosion of the composite shell structures of the gas exhaust ducts.

#### 2.2.2. Three-Point Bend Tests at Elevated Temperature

Three-point bending tests of the binder samples at temperatures of 23 °C and 100 °C were conducted according to GOST R 56810-2015 [26] on a Tinius Olsen h100ku (Tinius Olsen GmbH, Goethestr. 7b, 86161, Augsburg, Germany) in a temperature chamber, which provides heating to 300 °C. The accuracy of load measuring of the Tinius Olsen h100ku machine is ±0.5% in the range from 0.2 to 100% of the permissible load of the cell (100 kN). The crosshead has a resolution of 0.001 mm with an accuracy of 0.01 mm. To eliminate the influence of machine compliance, the displacement of the sample center point under load was also controlled by a mechanical indicator mounted under the sample. 

Samples were cut from undamaged portions of the plates tested for wear and then tested over a span of 43 mm. The tests determined modulus of elasticity and bending strength.

The three-point bending tests determined the cured sample deformation modulus at temperatures of 23 °C and 100 °C. The experimental values of elasticity modulus at the bending of the samples were determined at a 2 mm/min loading rate. The determination of the elasticity modulus was carried out under loading with two load steps.

When determining the elastic modulus in bending, the samples were preliminarily loaded with a concentrated force to the level of normal stresses of 5 MPa. Further loading was carried out, and the determination of the elastic modulus was carried out in the range of normal stresses of 10 MPa.

The samples were preliminarily held at elevated temperatures until they were completely warmed up to the test temperature. The temperature during the tests was maintained by a thermostat and controlled by two thermocouples. One thermocouple measured the temperature on the surface of the bent sample. The second thermocouple measured the temperature inside a control sample, located next to the test sample.

## 3. Results and Discussion

### 3.1. Solid Particle Erosion Testing

The test results for all the samples from all the compounds are shown in Table A1, Table A2, Table A3, Table A4, Table A5, Table A6, Table A7, Table A8, Table A9, Table A10, Table A11, Table A12, Table A13, Table A14, Table A15, Table A16, Table A17, Table A18, Table A19, Table A20, Table A21, Table A22, Table A23 and Table A24 in Appendix A. A summary of the results for the tested resin types is given in Table 2, and the same data is shown graphically in Figure 5, Figure 6 and Figure 7. The temperature columns in the tables show, outside parentheses, the temperature at the start of the test, and in parentheses, the average temperature during one test (which differs from the initial temperature due to cooling of the samples during the test).

Statistical processing of the results was carried out in accordance with Section 4 and Annex 3 of standard GOST 14359-69* [27]. The coefficient of variation was determined by the formula $Vc=\frac{S_{v}}{\bar{VE}}$, where *S_v_* is the standard deviation and $\bar{VE}$ is the arithmetic mean of the volumes of material lost through erosion.

Rough errors were determined based on $\left|\bar{VE}\right.-\left.VE_{i\;}\right|$* > t_a_ × Sv*, where $VE_{i\;}$ is the volume of material lost due to erosion in each individual test, *t_a_* is the statistical criterion for the distribution of normalized deviations in a small sample, depending on the number of tests and confidence probability. The values of the normalized deviation distribution criterion in the small sample *t_a_* were determined according to the number of tests for a 0.95 confidence level. Values outside the confidence intervals (gross errors) were discarded (they are crossed out in the tables of Appendix A).

Figure 8 shows the temperature dependence of the wear of filled and unfilled compounds at an impingement angle of 90°. In all cases, the wear of filled compounds is significantly higher than that of unfilled ones. Maximum wear is observed for the compound filled with fly-ash 50% (EP-FA50)—at 23 °C and 100 °C it is about 7–7.5 times higher than unfilled compound wear—at 180 °C the difference is 10 times higher.

EP-FA25 at 23 °C showed wear, coinciding with unfilled binders; the wear rate increased linearly with increasing temperature, in contrast to all other compounds studied. This result could be explained by the non-uniform distribution of filler particles over the sample thickness, which is shown below in Section 3.2. In other words, at the initial stage of erosion, mainly low-filled layers were subjected to wear, so the result is close to the unfilled compound.

A typical feature for the filled and unfilled compounds at an impingement angle of 90° (except EP-FA25) was a significant reduction of wear at 100 °C, which does not exceed the glass transition temperature. For the unfilled compounds (batches Ep1.1, Ep1.2) the wear rate decreased by 1.5–2.5 times, for the filled EP-FA50 and EP-FM33 by 1.4 and 3.8 times, respectively. This is associated with an increase in elasticity and fracture limit when heated, without a significant reduction in strength (which is shown in Section 3.2).

At a 90° impingement angle and 180 °C (higher than glass transition temperature) all the compounds showed higher wear than at 100 °C, and in all cases (except EP-FM25) it did not exceed the wear rate at 23 °C. At 180 °C, the erosion pattern of the unfilled polymer surface changed and became significantly more uneven, even to the naked eye (Figure 9b).

Figure 10 shows the temperature dependence of the wear of filled and unfilled compounds at an impingement angle of 45°; at an impingement angle of 90° the wear of filled compounds is significantly higher than that of unfilled ones in all cases. At 23 °C, compared to EP, the wear of the filled compounds is 2.0 times higher for EP-FA25, 2.3 times higher for EP-FM33, and 2.75 times higher for EP-FM50.

At an impingement angle of 45° and increasing temperature, the filled binders (in contrast to an impingement angle 90°) showed a different trend in wear rate under the same conditions. The maximum-filled EP-FA50 compound showed about the same wear rate at 23 °C and 100 °C, and at 180 °C the wear rate increased by 30%. EP-FA25, at a temperature of 100 °C, showed an increase of wear by 20%, and at a temperature of 180 °C there was a decrease of the wear by about 30%. EP-FM33 (like the unfilled EP compound) showed a linear decrease in wear with increasing temperature.

The surface erosion pattern of the unfilled and filled sample at an impingement angle of 45° is shown in Figure 11.

The wear dependences on temperature for filled compounds can be explained by the structural features of samples, caused by properties of the filler, the amount and its distribution in the thickness of the sample. To clarify this question, it will be necessary to investigate wear at smaller temperature steps and to model the structure of the hardened filled compounds and the impact of abrasive particles on them.

The most clearly seen dependences of wear on temperature (at impingement angles of 45° and 90°) are observed for unfilled EP (batches EP1, EP2), for clarity they are shown separately in Figure 12. At 45°, a clearly visible inversely proportional linear dependence of wear on temperature is observed. At 100 °C, the wear, compared to 23 °C, decreased by 1.4 times (for both batches of the binder), and at 180 °C by 2.8–3.2 times.

At an impingement angle of 90°, the temperature dependence of wear is non-linear: when heated from 23 °C to 100 °C, the degree of wear decreases and when heated to 180 °C, it increases slightly but remains lower than at 23 °C. At 180 °C, the wear of the unfilled samples at impingement angles of 45° and 90° is almost the same.

According to [18], the maximum intensity of solid particle erosion at an impingement angle of 90° is typical for brittle materials, and at an angle of 20° for ductile materials, i.e., with a decrease of the brittleness of material, the degree of wear should increase at an impingement angle of 90°. The material, at which the maximum degree of wear is observed at 45° is called semi-ductile. In our case such a simplified definition is not quite correct, because, as the experiments showed, with an increase of temperature from 23 °C to 180 °C, the difference of wear intensity between impingement angles 45° and 90° decreases, i.e., the brittleness contribution to the wear should increase and the ductility contribution should decrease, which is not true for the unfilled EP, because its brittleness decreases, and ductility increases with heating.

### 3.2. Three-Point Bending Test

The mechanical characteristics were determined to estimate their correlation with the wear rate and to obtain data that can be used in the design of composite structures.

Three-point bending tests to determine the elastic modulus and the bending strength were performed on beam samples cut from plates that were previously tested for solid particle erosion, and the surface of the samples was machined with an abrasive disk.

The unfilled EP samples were divided into two groups, one group was tested at 23 °C and the other at 100 °C, according to the method described in Section 2.

After the wear tests, only a small number of filled bar samples of EP-FA50 and EP-FA25 binders were produced, so they were not divided into groups and were tested only at 23 °C.

The bending test results for all samples are shown in Table A25, Table A26, Table A27, Table A28 and Table A29 of Appendix B, and the summary results for the different sample types are shown in Table 3.

When tested at 100 °C, the maximum deflection of the samples was limited to 7 mm, but not all samples were able to be broken due to the increase in elasticity (ultimate strain increases) because of heating. Since the glass transition temperature was not reached, the samples remained stiff enough to determine the modulus of elasticity in the stress range of 5–10 MPa. 

The modulus of elasticity at 100 °C decreased by 18% compared to at a temperature of 23 °C. The level of ultimate stresses in the samples which broke down was slightly higher than the ultimate stresses in the samples tested at 23 °C. After testing and cooling, the unbroken samples kept their deformed state at room temperature and recovered their original straight shape after heating to the glass transition temperature. A test sample that did not fracture is shown in Figure 13.

The average values of the elastic modulus and flexural strength of the filled compounds (EP-FA25, EP-FM33, EP-FA50) at 23 °C are shown in Table 3. The results show a significant increase in the modulus of elasticity when compared to the unfilled compounds (1.42 times for EP-FA25, 1.69 times for EP-FM33, 2.35 times for EP-FA50), with a smaller increase in strength (1.03 times for EP-FA25, 1.2 times for EP-FM33 and 1.14 times for EP-FA50).

Figure 14 shows enlarged cross sections of EP-FA25 (a) and EP-FA50 (b). The figure shows that the sample filled at 50% by weight has a more homogeneous structure (the filler is more evenly distributed across the cross-section). The sample filled at 25% by weight has the largest particles of filler partially sedimented during curing at a high temperature and the liquefaction of the epoxy resin. Despite this, all the filled samples showed acceptable statistical variation in mechanical properties. Figure 15 shows the experimental dependence of the modulus of elasticity of unfilled and filled compounds on their density.

The correlation of wear values at impingement angles of 45° and 90° with elasticity modulus, strength, and density of the samples was evaluated using Microsoft Excel. The correlation coefficient between the sets of values *X* and *Y* is determined by:$$Correl\left(X,Y\right)=\frac{\sum \left(x-\bar{x}\right)(y-\bar{y})}{\sqrt{\;\sum {\left(x-\bar{x}\right)}^{2}\sum {\left(y-\bar{y}\right)}^{2}}}.$$

If the correlation coefficient is close to 1 or −1, then there is a significant direct or inverse correlation between the sets of values. The results of determining the correlation coefficient are presented in Table 4.

The results show that wear at impingement angles of 45° and 90° has a strong direct correlation with the modulus of elasticity and density, and a weak correlation with the flexural strength of the materials. In this case, the data set for determining the correlation included mechanical characteristics of the compound filling and the test temperature.

The following generalizing conclusions can be made based on the results of the experimental studies:At temperatures from 23 °C and 100 °C, the intensity of solid particle erosion of unfilled hot-cured EP at an impingement angle of 45° is significantly higher than at 90°. When the temperature is increased to 180 °C, the wear intensities become almost the same.Gas abrasion of unfilled EP decreases with increasing temperature from 23 °C to 180 °C. At an impingement angle of 45°, the dependence is linear, at an impingement angle of 90°, the dependence is non-linear. The decrease of the wear intensity is probably caused by the increase of ultimate strain as a result of heating.The gas abrasion of the examined filled epoxy compounds (EP-FA25, EP-FA50, EP-FM33) is much higher, than unfilled EP at impingement angles of 45° and 90°. For instance, the wear of EP-FA50 at a 90° impingement angle at 23 °C and 100 °C is 7–7.5 times higher than EP, this difference increases up to 10 times when the temperature is increased to 180 °C. The filled compounds have shown differently directed dependences of intensity of wear on temperature (especially at an impingement angle of 45°), the character of the dependence is probably defined by the degree of filling and structural features of the samples.The wear intensity of the compounds at impingement angles of 45° and 90° has a strong direct correlation with the density and modulus of elasticity and has a weak correlation with the bending strength of the materials.

## 4. Conclusions

Increasing the temperature has a positive effect on the resistance to solid particle erosion (especially for the unfilled binder), which is associated with an increase in the fracture limit. At heating up to 100 °C wear resistance of unfilled epoxy binder increased by 1.5–2.5 times in comparison with resistance at room temperature. At 180 °C, higher than the glass transition temperature, resistance to gas abrasion of the EP also increased, but at an impingement angle of 90° the character of surface erosion changed, it became significantly more inhomogeneous.

Filled compounds showed lower resistance to solid particle erosion, in comparison with unfilled EP, therefore it is reasonable to use them in load bearing layers of structures to increase their rigidity, and unfilled ones in inner protective layers, directly exposed to abrasive impact.

The data on solid particle erosion of filled and unfilled polymeric epoxy compounds can be used to estimate the operational wear of composite shell structures of gas exhaust ducts with an inner gel coat. It is necessary to note that the tests applied on solid particle erosion are accelerated and the speed of abrasive particles is higher than at real operating conditions, therefore for use of the data for forecasting of the wear of real structures extrapolation of the experimental data on exhaust gases with lower speed is necessary.

For the practical application of the materials considered here, it is necessary to investigate the influence on resistance to solid particle erosion of such factors as the influence of chemically aggressive substances, thermal ageing, etc.

## Figures and Tables

**Figure 1 polymers-15-00001-f001:**
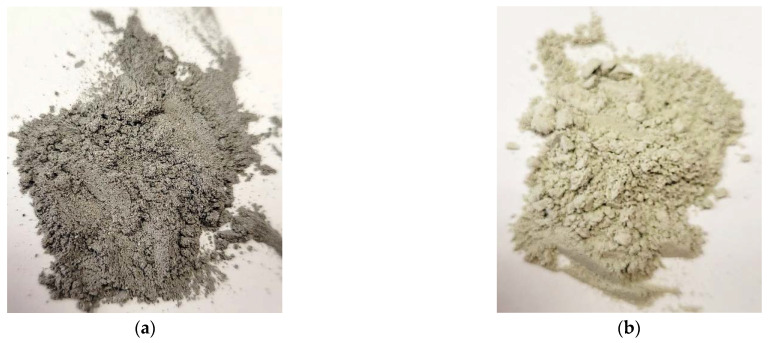
The fillers used: (**a**) Fly ash from the SDPP; (**b**) Backfill material obtained from the mineral processing waste of the copper-mining plant.

**Figure 2 polymers-15-00001-f002:**
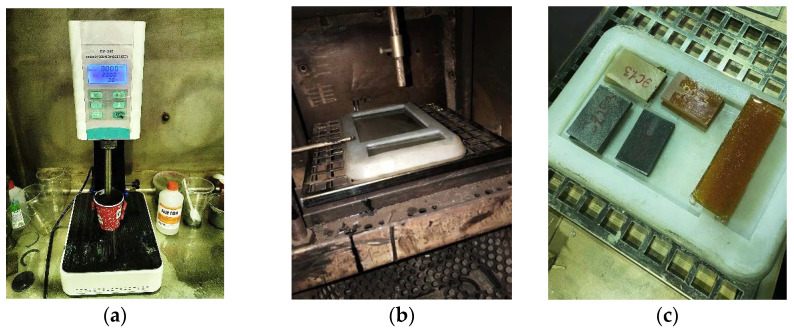
Sample preparation: (**a**) mixing in a homogenizer; (**b**) curing in a laboratory oven in a silicone mold; (**c**) prepared samples.

**Figure 3 polymers-15-00001-f003:**
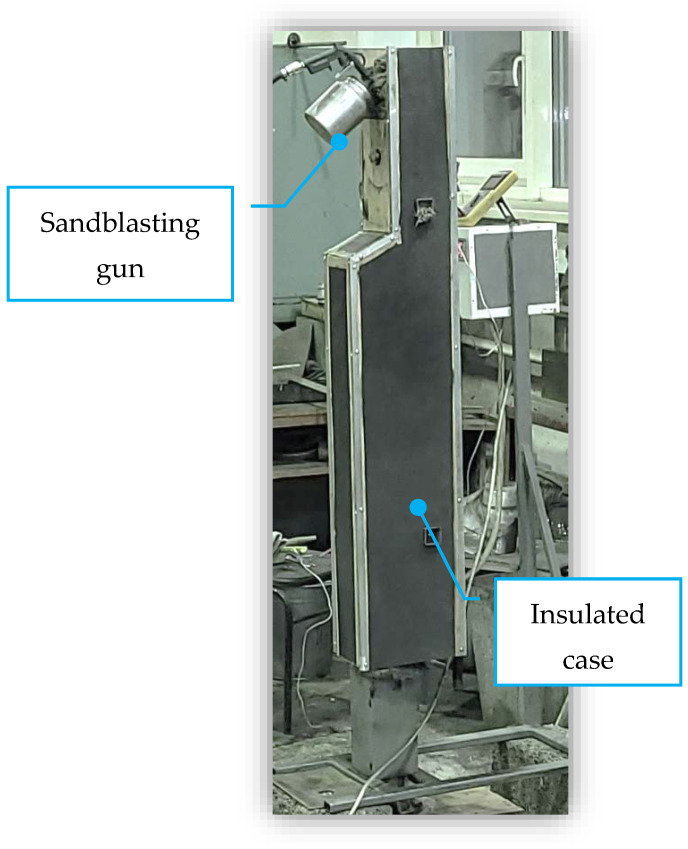
The experimental bench in the insulated exterior casing.

**Figure 4 polymers-15-00001-f004:**
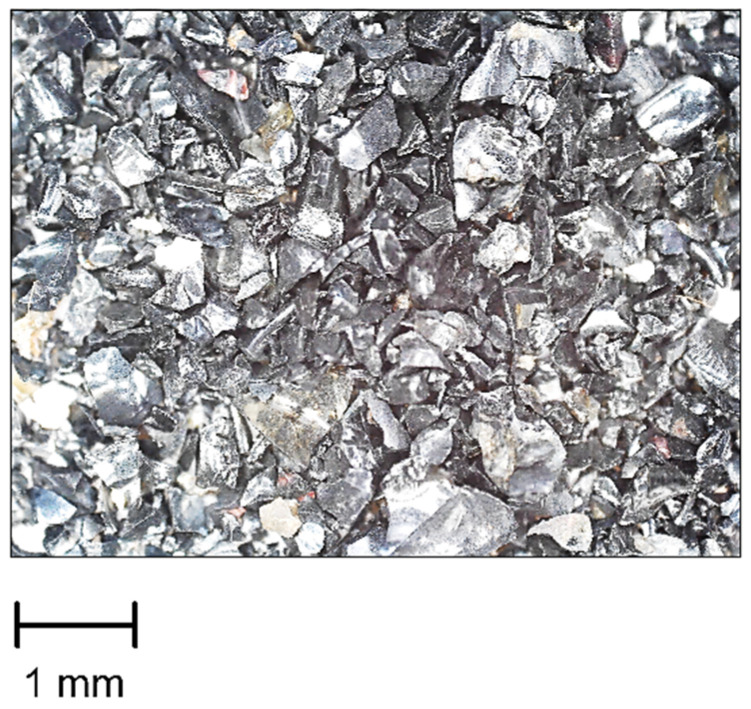
The sharp angular-shaped grains of 0.125–0.63 mm copper slag abrasive (scale of 10:1).

**Figure 5 polymers-15-00001-f005:**
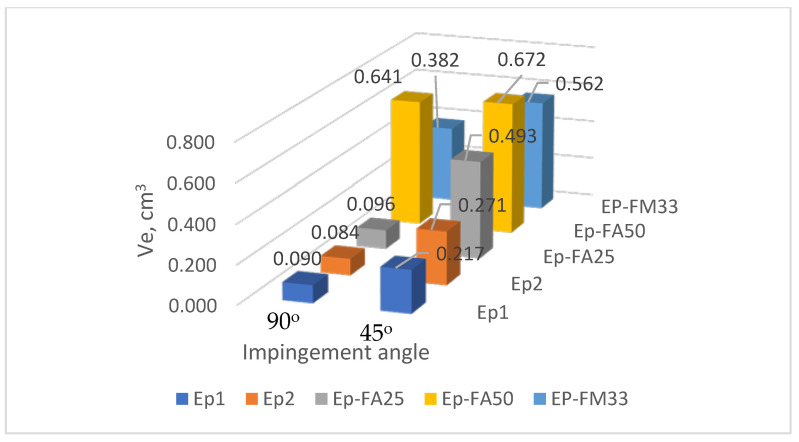
Compound wear at 23 °C at 90° and 45° impingement angles.

**Figure 6 polymers-15-00001-f006:**
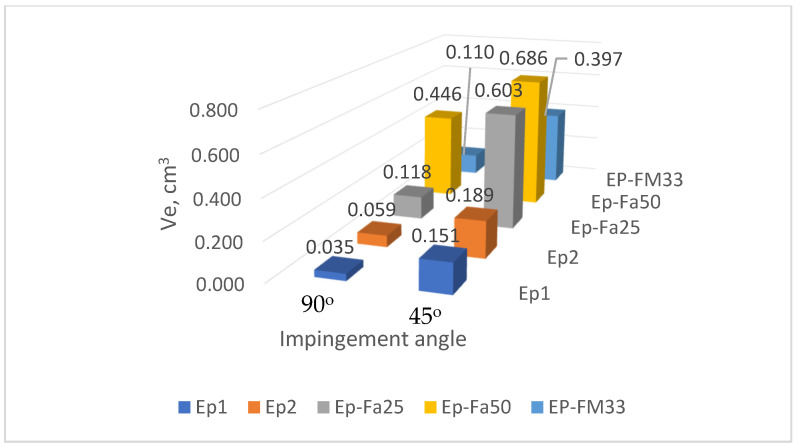
Compound wear at 100 °C at 90° and 45° impingement angles.

**Figure 7 polymers-15-00001-f007:**
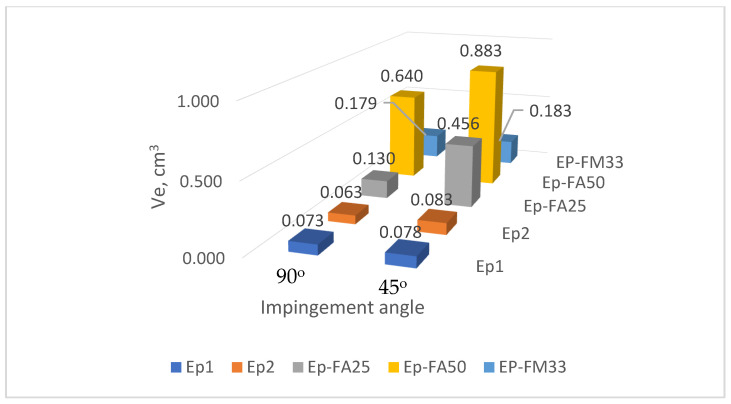
Compound wear at 180 °C at 90° and 45° impingement angles.

**Figure 8 polymers-15-00001-f008:**
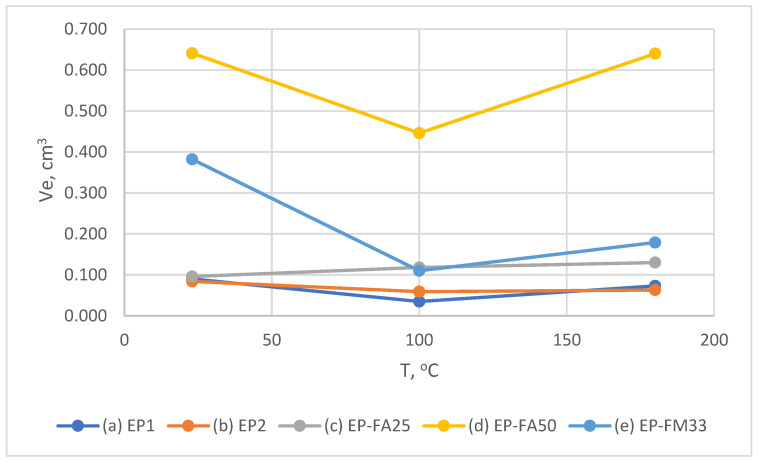
Dependence of the wear of unfilled and filled compounds on the temperature at an impingement angle of 90°.

**Figure 9 polymers-15-00001-f009:**
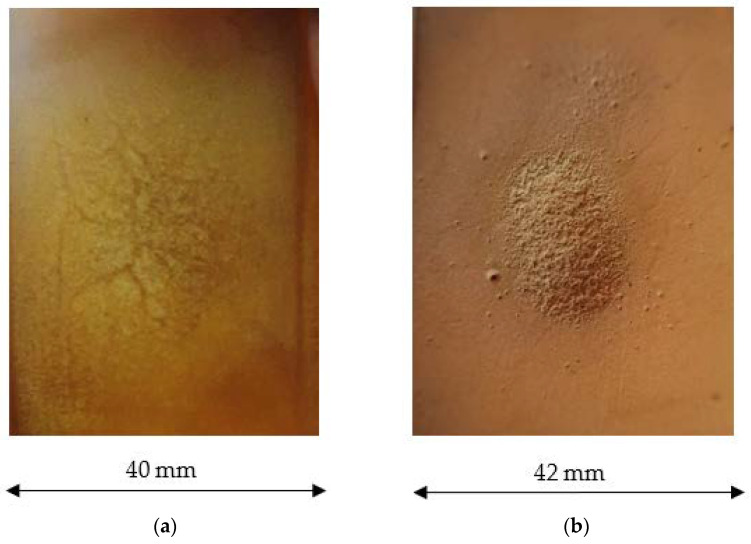
Wear of the unfilled epoxy compound surface at an impingement angle of 90° at temperatures: (**a**) up to 100 °C; (**b**) 180 °C.

**Figure 10 polymers-15-00001-f010:**
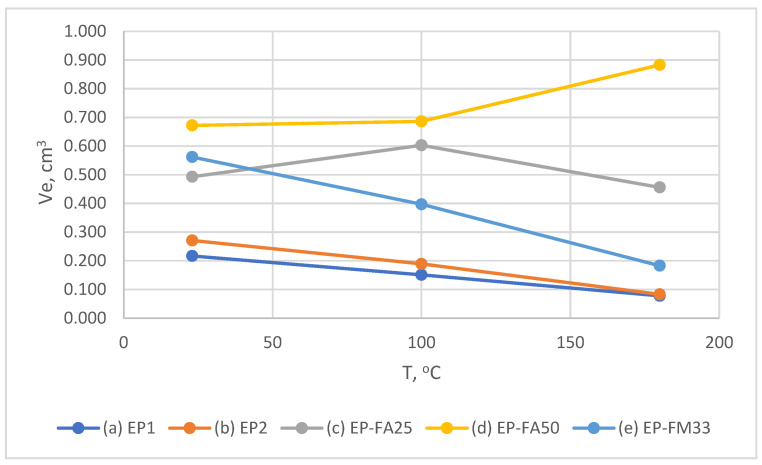
Dependence of the wear of unfilled and filled compounds on temperature at an impingement angle of 45°.

**Figure 11 polymers-15-00001-f011:**
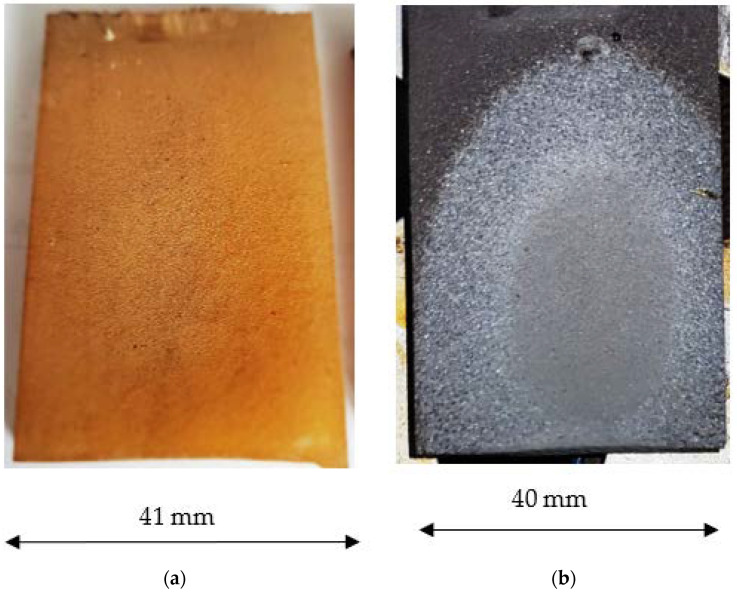
Surface erosion of unfilled (**a**) and filled (**b**) samples at an impingement angle of 45°.

**Figure 12 polymers-15-00001-f012:**
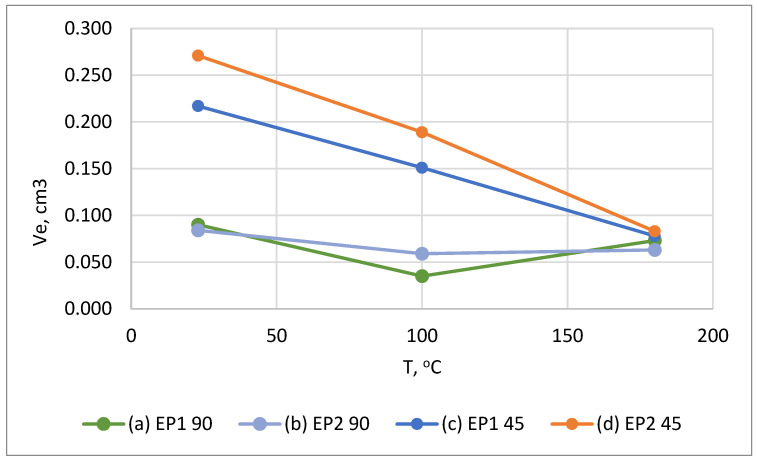
Dependence of wear of unfilled epoxy resin on temperature at impingement angles of 45° and 90°.

**Figure 13 polymers-15-00001-f013:**
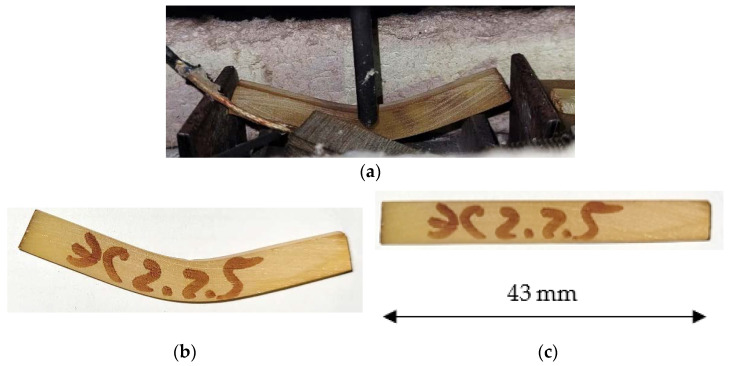
An unfilled epoxy binder sample: (**a**) during bending tests at 100 °C; (**b**) after removal of load and cooling in the deformed state; (**c**) after heating to the glass transition temperature and returning to the original shape.

**Figure 14 polymers-15-00001-f014:**
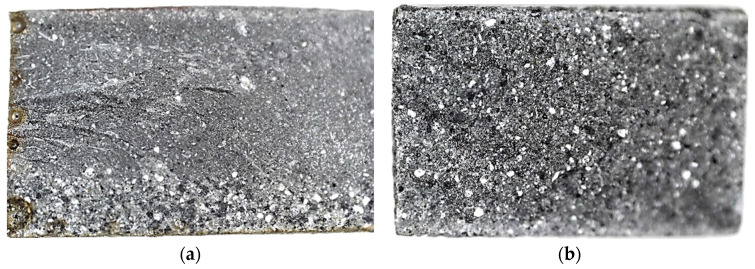
Cross-sections of filled samples: (**a**) EP-FA25; (**b**) EP-FA50.

**Figure 15 polymers-15-00001-f015:**
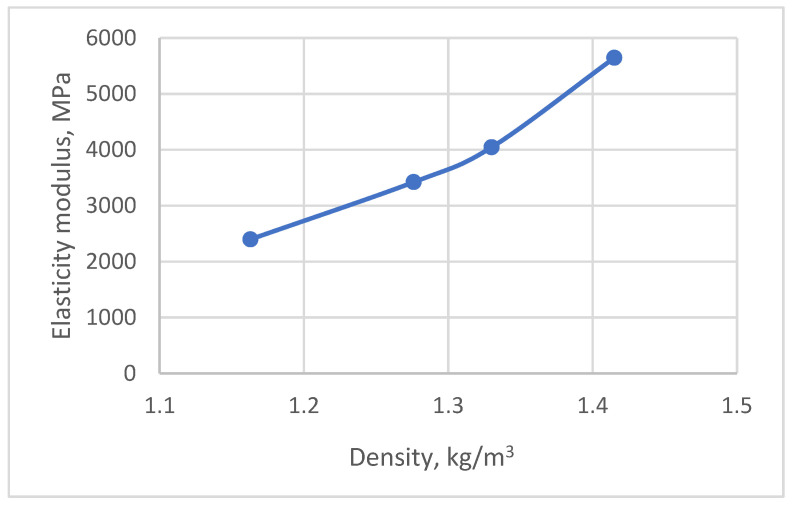
Dependence of elastic modulus on compound density.

**Table 1 polymers-15-00001-t001:** Types of binders investigated.

№	Composition	Composition	Designation
1	Epoxy resin	Ker 828 52.5% + MTHPA 44.5% + Alkofen 3%	EP
2	0.25 w/w fly-ash filled epoxy resin	EP 50% + Fly-Ash 25%	EP-FA25
3	0.5 w/w fly-ash filled epoxy resin	EP 50% + Fly-Ash 50%	EP-FA50
4	0.33 w/w filling material filled epoxy resin	EP 67% + Filling material 33%	EP-FM33

**Table 2 polymers-15-00001-t002:** Summary table of unfilled epoxy binder (EP) wear test results.

Type of Compound	Batch	Test Number	Variation Coefficient (%)	Density (g/cm^3^)	Average VE (cm^3^)	Temperature (°C)	Impingement Angle
EP	Ep1.1	8	20.13	1.167	0.090	23	90
		10	29.68		0.035	100 (96.5)	
		6	11.94		0.073	180 (165)	
	Ep1.2	8	21.11	1.167	0.084	23	
		10	23.75		0.059	100 (97)	
		6	5.29		0.063	180 (165)	
	Ep2.1	8	15.43	1.158	0.217	23	45
		6	10.23		0.151	100 (99)	
		6	14.3		0.078	180 (168)	
	Ep2.2	7	29.2	1.158	0.271	23	
		6	7.25		0.189	100 (101)	
		6	7.92		0.083	180 (173)	
EP-FA25	EP-FA25	6	17.9	1.276	0.096	23	90
		6	23.84		0.118	100 (98)	
		6	29.03		0.130	180 (173)	
	EP-FA25	6	7.35	1.276	0.493	23	45
		6	5.27		0.603	100 (101)	
		6	3.28		0.456	180 (174)	
EP-FA50	EP-FA50	6	15.31	1.415	0.641	23	90
		6	4.76		0.446	100 (100)	
		6	13.8		0.640	180 (172.5)	
	EP-FA50	6	2.74	1.415	0.672	23	45
		6	6.61		0.686	100 (99.5)	
		6	7.95		0.883	180 (175)	
EP-FM33	EP-FM33	13	29.15	1.330	0.382	23	90
		9	47.57		0.110	100 (96.5)	
		7	33.11		0.179	180 (173)	
	EP-FM33	13	33.74	1.330	0.562	23	45
		9	14.68		0.397	100 (96.7)	
		9	32.41		0.183	180 (166)	

**Table 3 polymers-15-00001-t003:** Summary of bend and wear test results.

Type of Compound	Temperature (°C)	Elastic Modulus (MPa)	Strength (MPa)	Average Erosion 90° (cm^3^)	Average Erosion 45° (cm^3^)	Density (g/cm^3^)	Modulus Variation Coefficient (%)	Strength Variation Coefficient (%)	Erosion 90° Variation Coefficient (%)	Erosion 45° Variation Coefficient (%)
EP	23	2398	60.7	0.087	0.244	1.163	12.9	36.6	20.62	22.32
EP	100	1970	71	0.047	0.170	1.163	28.1	11.0	26.18	8.74
EP-FA25	23	3421	62.4	0.096	0.493	1.276	11.4	27.2	17.9	7.35
EP-FA50	23	5645	69	0.641	0.672	1.415	8.1	8.6	15.31	2.74
EP-FM33	23	4045	73	0.382	0.562	1.33	17.1	19.5	29.15	33.74

**Table 4 polymers-15-00001-t004:** Correlation coefficients between physical and mechanical parameters and wear at different impingement angles.

Parameters	Modulus of Elasticity/ Erosion 90°	Modulus of Elasticity/ Erosion 45°	Strength/ Erosion 90°	Strength/ Erosion 45°	Density/ Erosion 90°	Density/ Erosion 45°
Correlation coefficient	0.949	0.965	0.365	0.044	0.900	0.988

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
