# Peer review of "Solid Particle Erosion of Filled and Unfilled Epoxy Resin at Room and Elevated Temperatures"

_polymers, 2022, doi:10.3390/polym15010001_

Round 1

Reviewer 1 Report

This is a interesting work dealing with the issue of  solid particle erosion of 

pure and filled hot-curing epoxy resin. This paper discusses in detail and the structure is reasonable. However, some questions need to be answered to make this paper more comprehensive.

1. Please supplement the application scenario and significance of this study.

2. There are many kinds of epoxy resin, and there are also many kinds of composite materials composed of different inorganic spices. What is the scope of application of this conclusion? Can we discuss and speculate. For example, bisphenol A epoxy and alumina composite.

3. Please supplement the subtitle of Figure 4.

4. Why is the label in Figure 13 red?

5. The conclusion of this paper should be more refined

Author Response

Dear reviewer, thank you for your interest to our work and your helpful comments. We have tried to take them fully into account and correct our work.

1. Please supplement the application scenario and significance of this study.

Revised the introduction, added a more detailed description of the possible practical application of the results of research in the introduction (lines 33-48). Also added Appendix C, which gave examples of designs of gas exhaust ducts, for the prediction of wear, which can be applied the results of the work.

2. There are many kinds of epoxy resin, and there are also many kinds of composite materials composed of different inorganic spices. What is the scope of application of this conclusion? Can we discuss and speculate. For example, bisphenol A epoxy and alumina composite.

We extended the description of the epoxy binder used and the dispersed fillers used. Compared to epoxy-alumina composites, the main advantage is the lower cost, because we used industrial waste, while solving the problem of their disposal (lines 55-67).

3. Please supplement the subtitle of Figure 4.

Supplemented the description of Figure 4.

4. Why is the label in Figure 13 red?

We corrected this mistake.

5. The conclusion of this paper should be more refined

The conclusion has been revised (lines 454-473) to be more compact

Reviewer 2 Report

This work presents an experimental study on the erosion of pure and filled hot-curing epoxy resins. An exhaustive review of previous works is presented, allowing the reader to comprehend the contribution of this work, and the importance of the studied materials.

English must be corrected, there are several sentences which are not clear.

Lines 185-192

Instead of a list of components, consider a description of how such components work, in order to provide a more fluid reading of this section.

Line 195

It would be better if “40…50 mm” is replaced by “40 mm to 50 mm”  or “40 mm – 50 mm”

The same for other dimension ranks in the text.

Figure 3.

I have a big concern with figure 3. Both images have been published before. It is important to consider different images, since this could be misinterpreted as self-plagiarism

Figure 4.

Please add a scale (with convenient dimensions)

Lines 230-242

Consider a description instead of a list. It would be easier to read

Lines 254-257

Temperature units are oddly displayed

Tables 2-5

Consider joining tables into a single one. Or consider formatting in a different way, since it is really hard to read as it is now.

Average VE and other columns: Add or discuss uncertainty calculation, specially for parameters which are resulting from a statistical analysis.

Figure 9, 10, 14.

Please add a scale (with convenient dimensions)

Conclusions

I recommend that you do not use the list format. Instead elaborate a more fluid discussion.

Author Response

Dear reviewer, thank you for your interest in our work and your helpful comments. We have tried to take them fully into account and correct them.

  1. Lines 185-192 Instead of a list of components, consider a description of how such components work, in order to provide a more fluid reading of this section.Made changes accordingly (lines 190-197 in the edited document).
  2. Line 195. It would be better if “40…50 mm” is replaced by “40 mm to 50 mm” or “40 mm – 50 mm”. The same for other dimension ranks in the text.

         Made appropriate changes to the text of the document.

3. Figure 3. I have a big concern with figure 3. Both images have been published before. It is important to consider different images, since this could be misinterpreted as self-plagiarism

 Indeed, this photo of our machine was previously published in our previous work. At first we decided that since we are using this equipment, it is possible to give its photo and the scheme according to which it was made. But we took your comment into account and changed the photo. We added a newer photo, it shows the bench after additional insulation.

4. Figure 4. Please add a scale (with convenient dimensions)

Added scale.

5. Lines 230-242. Consider a description instead of a list. It would be easier to read

We made the appropriate changes (lines 227-235 in the edited document).

6. Lines 254-257. Temperature units are oddly displayed

 Corrected throughout the entire document.

7. Tables 2-5. Consider joining tables into a single one. Or consider formatting in a different way, since it is really hard to read as it is now.

Average VE and other columns: Add or discuss uncertainty calculation, specially for parameters which are resulting from a statistical analysis.

Combined the tables into one, added a description of the parameters of statistical processing of the results (lines 279-289).

8. Figure 9, 10, 14. Please add a scale (with convenient dimensions)

 Added scale.

9. I recommend that you do not use the list format. Instead elaborate a more fluid discussion.

Revised the conclusion (lines 454-473).